The diagnostic and prognostic value of UBE2T in intrahepatic cholangiocarcinoma

Yu Hua 1
Wang Han 1
Dong Wei 1
Cao Zhen-Ying 1
Li Rong 1
Yang Chao 1
Cong Wen-Ming 1
Dong Hui huidongwh@126.com 1
Jin Guang-Zhi jgzhi@hotmail.com 2
1 Department of Pathology, Eastern Hepatobiliary Surgery Hospital, The Second Military Medical University , Shanghai , China
2 Department of Oncology, Tongren Hospital, Shanghai Jiao Tong University School of Medicine , Shanghai , China
Albertini Maria Cristina
Electronic publication date: 2020 Jan 27
Publication date: 2020
Volume: 8
Electronic Location ID: e8454
Received 2019 Sep 20; Accepted 2019 Dec 23
Copyright: ©2020 Yu et al.
Copyright year: 2020
Copyright holder: Yu et al.
License: This is an open access article distributed under the terms of the Creative Commons Attribution License, which permits unrestricted use, distribution, reproduction and adaptation in any medium and for any purpose provided that it is properly attributed. For attribution, the original author(s), title, publication source (PeerJ) and either DOI or URL of the article must be cited.
License URL: https://creativecommons.org/licenses/by/4.0/

Keywords: Intrahepatic cholangiocarcinoma, Immunohistochemistry, Biomarker, Diagnosis, Recurrence, Prognosis, UBE2T

Funding: National Natural Science Foundation of China 81472278 81472769 81972574 Mengchao Youth Talent Development Program We received funding from the National Natural Science Foundation of China (Grant No. 81472278, 81472769, 81972574) and the Mengchao Youth Talent Development Program for this study. The funders had no role in study design, data collection and analysis, decision to publish, or preparation of the manuscript.

==============================
Background

Ubiquitin-conjugating enzyme E2T (UBE2T) is overexpressed in several types of malignancies. However, little is known about its diagnostic significance in intrahepatic cholangiocarcinoma (ICC) and other bile duct diseases or its prognostic value in ICC.

Methods

The expression levels of UBE2T in the intrahepatic bile duct (IHBD, N = 13), biliary intraepithelial neoplasia (BilIN; BilIN-1/2, N = 23; BilIN-3, N = 11), and ICC (N = 401) were examined by immunohistochemistry. The differential diagnostic and prognostic values were also assessed.

Results

The number of UBE2T-positive cells was significantly higher in ICC tissues than in nonmalignant tissues, including the IHBD, BilIN-1/2, and BilIN-3 tissues. Kaplan–Meier analysis showed that overexpression of UBE2T was correlated with a shorter time to recurrence (TTR) and overall survival (OS). The 5-year TTR rates in the high UBE2T and low UBE2T groups were 100% and 86.2%, respectively. The corresponding OS rates were 1.9% and 22.2%, respectively. High expression of UBE2T was an independent risk factor for both TTR (hazard ratio: 1.345; 95% confidence interval: 1.047,1.728) and OS (hazard ratio: 1.420; 95% confidence interval: 1.098,1.837).

Conclusions

UBE2T can assist in differentiating benign bile duct diseases from ICC, and high expression of UBE2T suggests a poor prognosis for ICC.

Introduction

Intrahepatic cholangiocarcinoma (ICC) is the second most frequent liver malignancy and originates from intrahepatic bile duct epithelial cells (Bridgewater et al., 2014). ICC accounts for approximately 10%–15% of primary liver cancers and is characterized by its poor prognosis, with a 5-year survival rate of 25–35% after surgical resection (De Jong et al., 2011; Wang et al., 2013). The main factor in the poor prognosis of ICC is the presence of distant metastasis at the time of diagnosis, which heavily affects the implementation and curative effect of therapy (Weber et al., 2015). For example, the 5-year OS rates for stage III and stage IV disease are 10% and 0%, respectively (Valle, 2010). Thus, special attention should be paid to novel available biomarkers for both differential diagnoses and prognostic predictions of ICC.

Ubiquitin-conjugating enzyme E2T (UBE2T, also called HSPC150), a member of the E2 family, participates with particular E3 ubiquitin ligase in degrading target substrates (Jentsch, 1992). UBE2T was initially found in a case of Fanconi anemia (Machida et al., 2006). It also plays a crucial role in cellular development (Lim, Song & Baek, 2016). Importantly, the UBE2T gene, located at 1q32.1, has been reported to be overexpressed in several malignant tumors, such as bladder cancer, hepatocellular carcinoma, myeloma, and renal cell carcinoma (Gong et al., 2016; Hao et al., 2019; Liu et al., 2017; Zhang et al., 2019). However, little is known about the correlation between UBE2T and ICC.

To better characterize the effects of UBE2T, we examined its expression in ICC, the intrahepatic bile duct (IHBD), and biliary intraepithelial neoplasia (BilIN). Then, we assessed its utility in the differential diagnosis of benign and malignant bile duct diseases. Finally, the prognostic impact of UBE2T on ICC patients was examined.

Materials & Methods

Patients

Thirteen IHBD, 23 BilIN-1/2, and 11 BilIN-3 tissues from April 2008 to November 2013 were obtained, and 401 ICC formalin-fixed paraffin-embedded (FFPE) specimens diagnosed at the Eastern Hepatobiliary Surgery Hospital, Shanghai, from July 2000 to December 2008 were included; all specimens were classified into the diagnostic or prognostic group. Two experienced pathologists reexamined the associated hematoxylin and eosin (HE)-stained slides containing FFPE tissues. The inclusion criteria were as follows: (1) pathological diagnosis according to the histological diagnostic criteria of the WHO; (2) no preoperative anti-cancer treatment; (3) partial hepatectomy with curative intent; and (4) the existence of complete data for both the UBE2T H-score and follow-up period. Laboratory tests were conducted on blood samples obtained before surgery. The tumor diameter and surgical margin were tested on the largest tumor by using gross pathologic specimens. Written informed consent was obtained from all patients. The study protocol was approved by the Research Ethics Committee of Eastern Hepatobiliary Surgery Hospital (EHBHKY2015-01-001).

Follow-up

The endpoints of this study were time to recurrence (TTR) and overall survival (OS). TTR was defined from the date of hepatic resection until the detection of tumor recurrence, death or last observation. OS was defined as the interval between surgery and death or last observation. The patients’ follow-up examinations were performed every 3 months during the first year after surgery and every 6 months thereafter. At each visit, the tests for liver functions, alpha fetoprotein (AFP), carcinoembryonic antigen (CEA) and carbohydrate antigen 19-9 (CA199) and an abdominal ultrasound were conducted. Contrast-enhanced computed tomography (CT) scanning or magnetic resonance imaging (MRI) was performed once every 3-6 months or when recurrence or metastasis was suspected. Follow-up data were collected until September 2014.

Data mining of the TCGA database

Data on the mRNA expression of UBE2T and associated clinical prognosis were obtained from the data library of The Cancer Genome Atlas (TCGA).

Construction of tissue microarrays and IHC

As previously reported, tissue microarray construction, IHC, and H-score measurements were performed for all the cases (Hirsch et al., 2003; Yu et al., 2019). The HE-stained slides of all patients were reviewed and identified, and then, the typical areas were premarked in the paraffin blocks. The marked areas of each block were punched by tissue cylinders with diameters of 1.5 mm. As in previous studies, the representativeness of a 0.6-mm core in the tissue microarray was equal to that of a larger slide for the optimization of standardized experimental conditions and for assessing focal and heterogeneous expression types (Anagnostou et al., 2010; Jones & Prasad, 2012), so the expression characteristics of a tissue specimen of this size were considered to be representative of the whole tissue slice. Then, the obtained tissues were incorporated into a recipient paraffin block afterwards. After the sections were sliced to 4-µm thickness, they were placed on slides coated with 3-aminopropyltriethoxysilane.

Paraffin sections were deparaffinized in xylene and rehydrated through decreasing concentrations of ethanol (100%, 95%, and 85% for 5 min each). Antigens were unmasked by microwave irradiation for 3 min in pH 6.0 citric buffer and cooled at room temperature for 60 min. Endogenous peroxidase activity was blocked by incubation in 3% H2O2/phosphate-buffered saline, and goat serum was used to block nonspecific binding sites. The primary antibody was as follows: rabbit polyclonal antibody to UBE2T (GTX106464; GeneTax, USA; 1:200 dilution). An EnVision Detection kit (GK500705; Gene Tech, Shanghai, China) was employed to visualize UBE2T. Tissue sections were counterstained with hematoxylin for 5 min. Negative control slides without primary antibody were generated for all assays.

The UBE2T-stained slides were scanned on a KFBIO KF-PRO-005-EX digital section scanner (Konfoong Biotech International Co., Ltd., Yuyao, China). Images were analyzed using the HALO 2.0 CytoNuclear Quantification algorithm (Indica Labs), a whole-slide imaging data analysis software program that measures and reports individual cell data that is represented as the percentage of positive cells per mm2 tissue. The H-score was used to express the measurement of the HALO software (Fig. S1) (Kargl et al., 2017). H − score=1×%cells1++2×%cells2++3×%cells3+.

X-tile analysis

The optimal cutoff point used for the survival analysis of the different UBE2T expression groups was calculated with X-tile software version 3.6.1 (Yale University School of Medicine, New Haven, CT, USA) (Camp, Dolled-Filhart & Rimm, 2004). UBE2T expression was represented as the H-score, and X-tile plots were used for optimization of cutoff points based on follow-up data. Statistical significance was evaluated by a standard log-rank method using the cutoff score derived from 401 ICC cases, with the P value obtained from a lookup table.

Statistical analysis

Continuous variables were compared using the Mann–Whitney test or Student’s t-test. The chi-squared test or Fisher’s exact test were used to analyze categorical variables when appropriate. Multiple sets of measurement data were compared by means of a multiple independent samples nonparametric test (Kruskal-Wallis test). Receiver operating characteristic (ROC) area under the curve (AUC) analysis showed the discriminatory power of the putative markers. The TTR and OS rates were calculated by the Kaplan–Meier method and log-rank test. Univariate and multivariate prognostic analyses were conducted using the Cox proportional hazards regression model. A P value less than 0.05 indicated statistical significance. All statistical analyses were performed using SPSS 24.0 software (SPSS, Chicago, IL, USA) and GraphPad Prism 8.0.1 (GraphPad Software, La Jolla, CA, USA).

Results

Expression differences in and prognostic impact of UBE2T on ICC based on the TCGA database

Based on the data obtained from the TCGA database, 9 cases of normal tissue and 36 cases of ICC were analyzed. The UBE2T mRNA expression level in the ICC tissues was obviously higher than that in the normal tissues (P < 0.0001, Fig. 1A). For the 28 ICC patients with follow-up data, when the median UBE2T expression value was used as the cutoff point, the survival analysis showed no difference in OS (P = 0.3679, Fig. 1B).

Figure 1 UBE2T levels in normal tissue and ICC (A) and its prognostic impact on ICC (B) based on the TCGA database.

UBE2T expression profiles of the IHBD, BilIN-1/2, BilIN-3, and ICC and its diagnostic value

The H-scores for 13, 23, 11, and 401 IHBD, BilIN-1/2, BilIN-3, and ICC tissues, respectively, were calculated for UBE2T expression-level comparisons. The immunohistochemical expression characteristics of UBE2T in the IHBD, BilIN-1/2, BilIN-3, and ICC tissues are shown in Fig. 2. The UBE2T expression level of the ICCs was significantly higher than that of the IHBD (P = 0.003), BilIN-1/2 (P < 0.001), and BilIN-3 tissues (P = 0.012) (Fig. 3A). Additionally, the ROC curve revealed that there was a strong discrimination between ICC and the IHBD (AUC = 0.782), BilIN-1/2 (AUC =0.774), and BilIN-3 tissues (AUC = 0.776) (Fig. 3). The baseline characteristics of the three latter groups are shown in Table S1.

Figure 2 Immunohistochemical expression characteristics of UBE2T in the IHBD (A), BilIN-1/2 (B), BilIN-3 (C), and ICC (D) (20×).

Figure 3 UBE2T expression in the IHBD, BilIN-1/2, BilIN-3, and ICC by immunohistochemistry (A).

ROC curve analysis of UBE2T for differential diagnosis (B: IHBD and ICC, (C): BilIN-1/2 and ICC, (D): BilIN-3 and ICC).

Clinicopathological characteristics of the ICC cohort

For the 401 patients with ICC, analysis by X-tile software was performed to assess the best cutoff point for the H-score of UBE2T. Using a standard log-rank method, with P values acquired from a lookup table for TTR and OS, we selected an H-score of <28.96 as the best cutoff point. A total of 288 and 113 patients were divided into low- and high-expression groups, respectively (Fig. S2). The baseline characteristics of the ICC patients are shown in Table 1. The median follow-up time was 42.1 months.

Table 1 Baseline characteristics of ICC patients with different expression of UBE2T (N = 401).

Characteristics	Low (N = 288)	High (N = 113)	P value	
Age, years	53.58 ± 11.14	52.82 ± 9.83	0.688	
Sex			0.618	
Male	211(73.3%)	80(70.8%)		
Female	77(26.7%)	33(29.2%)		
AFP, ng/mL	81.69 ± 236.65	112.83 ± 286.36	0.069	
CEA, ng/mL	14.17 ± 84.30	14.76 ± 94.78	0.920	
CA199, ng/mL	169.47 ± 274.57	140.65 ± 221.52	0.692	
TBIL, µmol/L	15.58 ± 15.57	24.66 ± 55.93	0.009	
ALB, g/L	42.15 ± 4.27	41.23 ± 4.67	0.144	
ALT, U/L	47.55 ± 67.75	43.06 ± 44.44	0.665	
GGT, U/L	108.22 ± 129.31	155.61 ± 256.95	0.060	
ALP, U/L	122.85 ± 96.98	130.06 ± 105.59	0.377	
HBsAg			0.085	
Positive	165(57.3%)	54(47.8%)		
Negative	123(42.7%)	59(52.2%)		
Tumor size, cm	6.15 ± 3.35	7.33 ± 3.56	0.002	
Tumor number			0.070	
Single	235(81.6%)	83(73.5%)		
Multiple	53(18.4%)	30(26.5%)		
Surgical margin			0.776	
<1 cm	213(74.0%)	82(72.6%)		
≥1 cm	75(26.0%)	31(27.4%)		
Liver cirrhosis			0.795	
Yes	75(26.0%)	28(24.8%)		
No	213(74.0%)	85(75.2%)		
MVI			0.018	
Positive	53(18.4%)	33(29.2%)		
Negative	235(81.6%)	80(70.8%)		
TNM			0.924	
I–II	241(83.7%)	95(84.1%)		
III–IV	47(16.3%)	18(15.9%)		
Notes.

ICC intrahepatic cholangiocarcinoma

AFP alpha fetoprotein

CEA carcinoembryonic antigen

CA199 carbohydrate antigen 19-9

TBIL total bilirubin

ALB albumin

ALT alanine aminotransferase

GGT gamma glutamyltransferase

ALP alkaline phosphatase

HBsAg hepatitis B surface antigen

MVI microvascular invasion, TNM tumor-node-metastasis

Impact of UBE2T on the clinical outcomes

Kaplan–Meier analysis was conducted to compare the prognoses of ICC patients with low or high expression of UBE2T. The results suggested that the high-expression group had a shorter TTR and OS than the low-expression group. The median TTRs of the high- and low-expression groups were 5.7 months and 8.5 months, respectively. The 1-, 3-, and 5-year TTR rates were 69.3%, 85.6%, and 100%, respectively, in the high-expression group and 56.2%, 72.5%, and 86.2%, respectively, in the low-expression group (P = 0.005, Fig. 4A). The median OS times of the high- and low-expression groups were 18.4 months and 24.2 months, respectively. The corresponding 1-, 3-, and 5-year OS rates were 64.4%, 25.7%, and 1.9%, respectively, in the high-expression group and 73.4%, 40.7%, and 22.2%, respectively, in the low-expression group (P < 0.001, Fig. 4B).

Figure 4 Cumulative incidences in the time to recurrence (A) and overall survival (B) curves comparisons for high and low expression of UBE2T.

Independent prognostic factors of TTR and OS

Univariable analysis by means of Cox proportional hazards regression revealed that AFP, CA199, tumor size, tumor number, liver cirrhosis, microvascular invasion (MVI), tumor-node-metastasis (TNM) staging, and UBE2T were associated with TTR (all P < 0.05, Table 2) and that AFP, CA199, albumin (ALB), alkaline phosphatase (ALP), tumor size, tumor number, MVI, TNM staging, and UBE2T were related to OS (all P < 0.05, Table 3). Multivariable analysis indicated that multiple tumor nodules, poorer TNM staging, and high UBE2T expression were independent risk factors for TTR (all P < 0.05, Table 2) and that higher CA199, lower ALB, larger tumor size, multiple tumor nodules, presence of MVI, poorer TNM staging, and high UBE2T expression were significant risk factors for OS (all P < 0.05, Table 3).

Table 2 Independent risk factors of time to recurrence.

Characteristics	Univariate	Multivariate	
	HR(95% CI)	P value	HR(95% CI)	P value	
Age, year	0.990(0.980,1.001)	0.067			
Sex: female vs. male	0.858(0.664,1.108)	0.240			
AFP, ng/mL	1.000(1.000,1.001)	0.040			
CEA, ng/mL	1.000(0.999,1.001)	0.922			
CA199, ng/mL	1.000(1.000,1.001)	0.024			
TBIL, µmol/L	0.999(0.995,1.003)	0.484			
ALB, g/L	0.996(0.970,1.022)	0.737			
ALT, U/L	1.001(0.999,1.002)	0.442			
GGT, U/L	1.000(1.000,1.001)	0.251			
ALP, U/L	1.001(1.000,1.002)	0.068			
HBsAg: positive vs. negative	0.831(0.663,1.043)	0.110			
Tumor size, cm	1.074(1.044,1.105)	<0.001			
Tumor number: multiple vs. single	1.943(1.483,2.545)	<0.001	1.867(1.423,2.449)	<0.001	
Surgical margin: <1 cm vs. ≥1 cm	1.299(0.999,1.689)	0.051			
Liver cirrhosis: positive vs. negative	0.690(0.525,0.906)	0.008			
MVI: positive vs. negative	1.363(1.035,1.794)	0.027			
TNM: III–IV vs. I–II	1.460(1.075,1.981)	0.015	1.447(1.066,1.964)	0.018	
UBE2T: high vs. low	1.422(1.108,1.825)	0.006	1.345(1.047,1.728)	0.020	
Notes.

AFP alpha fetoprotein

CEA carcinoembryonic antigen

CA199 carbohydrate antigen 19-9

TBIL total bilirubin

ALB albumin

ALT alanine aminotransferase

GGT gamma glutamyltransferase

ALP alkaline phosphatase

HBsAg hepatitis B surface antigen

MVI microvascular invasion

TNM tumor-node-metastasis

Table 3 Independent risk factors of overall survival.

Characteristics	Univariate	Multivariate	
	HR(95% CI)	P value	HR(95% CI)	P value	
Age, year	0.997(0.986,1.007)	0.529			
Sex: female vs. male	0.980(0.757,1.269)	0.878			
AFP, ng/mL	1.001(1.000,1.001)	0.011			
CEA, ng/mL	1.000(0.999,1.001)	0.886			
CA199, ng/mL	1.001(1.001,1.001)	<0.001	1.001(1.000,1.001)	0.002	
TBIL, µmol/L	1.000(0.997,1.003)	0.942			
ALB, g/L	0.953(0.928,0.979)	<0.001	0.965(0.939,0.992)	0.012	
ALT, U/L	1.001(0.999,1.003)	0.381			
GGT, U/L	1.000(1.000,1.001)	0.368			
ALP, U/L	1.001(1.000,1.002)	0.014			
HBsAg: positive vs. negative	0.844(0.670,1.063)	0.150			
Tumor size, cm	1.068(1.040,1.098)	<0.001	1.039(1.007,1.072)	0.015	
Tumor number: multiple vs. single	1.908(1.467,2.481)	<0.001	1.604(1.217,2.115)	0.001	
Surgical margin: <1 cm vs. ≥1 cm	1.273(0.976,1.660)	0.075			
Liver cirrhosis: positive vs. negative	0.900(0.693,1.169)	0.429			
MVI: positive vs. negative	1.725(1.330,2.238)	<0.001	1.345(1.026,1.763)	0.032	
TNM: III–IV vs. I–II	1.865(1.385,2.511)	<0.001	1.578(1.142,2.181)	0.006	
UBE2T: high vs. low	1.609(1.259,2.057)	<0.001	1.420(1.098,1.837)	0.008	
Notes.

AFP alpha fetoprotein

CEA carcinoembryonic antigen

CA199 carbohydrate antigen 19-9

TBIL total bilirubin

ALB albumin

ALT alanine aminotransferase

GGT gamma glutamyltransferase

ALP alkaline phosphatase

HBsAg hepatitis B surface antigen

MVI microvascular invasion

TNM tumor-node-metastasis

Discussion

It is notable that the incidence rate of ICC, a rare form of liver cancer, has been rising globally over the past twenty years, which may reflect both a true increase and more accurate diagnosis of the disease (Massarweh & El-Serag, 2017; Zhang et al., 2016). Additionally, ICC is still a poorly understood malignancy that is strongly characterized by its poor prognosis (Bagante et al., 2017; Beal et al., 2018). Although scholars have revealed that some conventional medical interventions, such as adjuvant chemotherapy, antiviral therapy, and anatomical resection, might improve the prognosis of ICC patients to some extent, the 5-year OS rate is still only 40% (Lei et al., 2018; Schweitzer et al., 2017; Si et al., 2019). Thus, molecular biomarkers that are precise and targeted have the potential to become new methods for the accurate diagnosis and individualized treatment of ICC (Sirica et al., 2019).

With the development of molecular pathology, an increasing number of biomarkers have been associated with the differential diagnosis and prognostic prediction of ICC (Rahnemai-Azar et al., 2017). Regarding the differential diagnosis, Matsushima et al. (2015) reported that Sex-determining region Y-box9 (Sox9) is overexpressed and associated with the carcinogenesis of ICC. Decreased Sox9 expression may be related to the early stage of ICC. Another Japanese multicenter study confirmed that Wisteria floribunda agglutinin (WFA)-sialylated mucin core polypeptide 1 (MUC1) is a useful biomarker of benign biliary tract diseases and ICC (Shoda et al., 2017). Regarding the prediction of recurrence and prognosis, the loss of Secreted frizzled-related protein-1 (SFRP1) was shown to indicate poor disease-free survival and OS for ICC patients through its effects on the Wnt-β-catenin pathway (Davaadorj et al., 2017). Similarly, the findings of a study analyzing 56 cases of ICC suggested that high Bim expression in tumors was correlated with better prognosis through inhibition of tumor cell proliferation and metastatic ability (Zhang et al., 2018). However, although the molecular profile of ICC has been reviewed and emphasized, the range of application is still smaller than that of morphological subclassification (Liau et al., 2014; Oliveira et al., 2017), which compels us to explore additional strong biomarkers for both the differential diagnosis and prognostic prediction of ICC.

UBE2T was initially reported to be the ubiquitin-conjugating enzyme in the Fanconi anemia pathway and had a self-inactivation mechanism that could be essential for negative regulation of the Fanconi anemia pathway (Zhang, Zhou & Huang, 2007). Interestingly, subsequent studies showed that UBE2T downregulation inhibited the proliferation, migration, and invasion of many types of tumor cells and that its depletion significantly suppressed tumor formation and the metastasis of malignancies. For instance, Wen M et al. showed that increased UBE2T expression was associated with oncogenic properties in human prostate cancer (Wen et al., 2015). Wang Y et al. reported that knockdown of UBE2T inhibited the proliferation, migration, and invasion of osteosarcoma cells (Wang et al., 2016). Perez-Pena et al. (2017) indicated that UBE2T was amplified in non-small cell lung adenocarcinomas and linked to recurrence after surgery. These findings may be regarded as evidence supporting the role of UBE2T as a cancer-promoting gene.

Considering the rapid tumor progression of ICC and the prominent role of UBE2T in tumor formation and development, in the present study, we performed corresponding research to investigate the effects of UBE2T on the diagnosis and prognosis of ICC. Based on the TCGA database, we confirmed that UBE2T was more highly expressed in the tumor areas than in the normal tissues of ICC patients. There was no significant difference in the prognoses of the high- and low-expression groups of ICC patients, but this may be attributed to the small number of cases and unsuitable cutoff points. Therefore, we continued to validate our hypothesis by evaluating the patients admitted to our center. First, we focused on the value of UBE2T for the differential diagnosis of ICC and other bile duct diseases. It was remarkable that the expression level of UBE2T was higher in ICC than in the IHBD, BilIN-1/2, or BilIN-3 and that UBE2T could be used as a discriminating marker between ICC and the other three tissue types. Thus, we suppose that UBE2T is a potential practical protein molecule for differential diagnoses. Second, we explored the prognostic influence of UBE2T in ICC. The X-tile software divided the ICC patients as 288 cases in the low-expression group and 113 cases in the high-expression group, and the clinical outcomes were obviously undesirable in the high-expression group. The 5-year TTR rates were 100% and 86.2% for the low- and high-expression groups, respectively, and the OS rates were 1.9% and 22.2%, respectively, for these two groups. In addition, both the univariate and multivariate Cox regression analyses revealed that UBE2T was an independent risk factor for TTR and OS. Specifically, UBE2T plays an important role in the malignant transformation and tumor development of ICC. Additionally, the results of our Cox regression analysis showed that many tumor behavior parameters, including tumor number, tumor diameter, TNM staging and liver function parameters such as ALB, are statistically significantly associated with TTR or OS. Interestingly, tumor number was a risk factor for both TTR and OS regardless of TNM staging; we hypothesize that different clonal origin patterns (intrahepatic metastasis and multicentric origin) exist in multinodular tumors and have more complex prognostic implications. Thus, we recommend that UBE2T and other risk factors be considered together in the prognostic prediction of ICC.

To our knowledge, this is the first study to demonstrate the correlation between UBE2T and ICC in a large cohort of patients. However, a prospective, multicenter cohort study is still necessary to confirm our conclusion. Furthermore, the specific molecular mechanism of UBE2T in the biological behaviors of ICC requires future investigation.

Conclusions

UBE2T is a useful biomarker for the differential diagnosis of ICC. High UBE2T expression in ICC tissues is an independent indication of a poor prognosis. UBE2T is a potential drug target for molecular targeted therapy of ICC in the future.

Supplemental Information

Supplemental Information 1 Supplemental figure and table

Click here for additional data file.

Supplemental Information 2 UBE2T microarray and baseline data

Click here for additional data file.

Supplemental Information 3 Miame Checklist

Click here for additional data file.

Additional Information and Declarations

Competing Interests

Author Contributions

Human Ethics

Data Availability

The authors declare there are no competing interests.

Hua Yu performed the experiments, analyzed the data, prepared figures and/or tables, authored or reviewed drafts of the paper, and approved the final draft.

Han Wang performed the experiments, analyzed the data, prepared figures and/or tables, authored or reviewed drafts of the paper, and approved the final draft.

Wei Dong analyzed the data, prepared figures and/or tables, and approved the final draft.

Zhen-Ying Cao performed the experiments, prepared figures and/or tables, and approved the final draft.

Rong Li analyzed the data, prepared figures and/or tables, and approved the final draft.

Chao Yang analyzed the data, prepared figures and/or tables, and approved the final draft.

Wen-Ming Cong conceived and designed the experiments, authored or reviewed drafts of the paper, administrative support, and approved the final draft.

Hui Dong conceived and designed the experiments, authored or reviewed drafts of the paper, and approved the final draft.

Guang-Zhi Jin conceived and designed the experiments, authored or reviewed drafts of the paper, and approved the final draft.

The following information was supplied relating to ethical approvals (i.e., approving body and any reference numbers):

The study protocol was approved by the Research Ethics Committee of Eastern Hepatobiliary Surgery Hospital (EHBHKY2015-01-001).

The following information was supplied regarding data availability:

The raw data is available as a Supplementary Files.

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
