# Peer review of "The diagnostic and prognostic value of UBE2T in intrahepatic cholangiocarcinoma"

_PeerJ, doi:10.7717/peerj.8454_

## Round 0.1 · original submission · Minor Revisions

Following the reviewers suggestions will improve the quality of the manuscript.

Reviewer 1 ·

Basic reporting

1. In “Abstract” section, the description of results should be more clear.
2. The part of “Discussion” is recommended to add some description and literature references about the progress of biomarker of ICC.

Experimental design

The experimental design is appropriate. However, according to WHO classification, LGIN should be described as BilIN-1 and -2. HGIN should be replaced as BilIN-3. Due to few cases included and similar biological behaviors, BilIN-1 and -2 can be combined as one group, but diagnosis name must be corrected.

Validity of the findings

1. In Table 1-3, the TNM staging of ICC patients should be showed in paper.
2. In Supplementary Table 1, the P value of comparison of IHBD, BilIN-1/2, and BilIN-3 should be provided.
3. In Figure 4, the number of risks should be revealed in the Figure of Kaplan-Meier analysis.

Additional comments

This is an interesting paper evaluating UBE2T expression as a diagnostic and prognostic biomarker for ICC which is targeting to give a better direction to clinicians once they need to take a decision concerning personalized therapy. The study is well designed and data is reliable. However, some issues should be addressed.

Annotated reviews are not available for download in order to protect the identity of reviewers who chose to remain anonymous.

Reviewer 2 ·

Basic reporting

The study investigated the UBE2T protein expression in intrahepatic cholangiocarcinoma (ICC) by tissue microarray (TMA) and concluded that overexpression of UBE2T in ICC is associated with poor outcome including early time to recurrence (TTR) and short overall survival (OS).The study is well-designed and well-analyzed. The conclusion appears to be well-founded and recommend to be accepted for publication. However, minor issues still need to be addressed.

Experimental design

1. Large respective series study of intrahepatic cholangiocarcinoma (401 cases)
2. Well-defined inclusion criteria with well-documented post-surgical follow up with ultrasound or radiography (CT)
3. Immunostains on TMA were digitally scanned for H-score and results were reviewed by two expert pathologists
4. In recognizing the shortfalls of the TCGA database, authors increased the sample size with optimal cut-point for data selection and analysis

Validity of the findings

The study of UBE2T expression in ICC appears to be well-conducted with appropriate controls and data analysis. The conclusion that UBE2T potentially a important prognostic factor is acceptable.

Additional comments

1. Was UBE2T protein expression homogenous on immunostain? If not, did authors compare the UBE2T expression on TMA with whole slide?
2. ~20% ICC showed multiple tumors, however, authors did not discuss whether they were true multifocality or vascular invasion or intrahepatic metastasis, and although sometime it’s very difficult to separate them.
3. Figure 2 showed increased protein expression of UBE2T from low grade dysplasia to invasive ICC, however, Figure 3 A showed similar level of expression of UBE2T from nondysplastic epithelium to high grade dysplasia (HGIN), please explain.
4. Since it appears that UBE2T primarily expressed in ICC, the suggestion of UBE2T can be used as a marker for early detection will be doubtful (see line 211)
5. Margin status of surgical resection will have huge impact on the tumor recurrence and overall survival, the authors did not mention it in the paper
6. Although the paper is overall well-composed, there’s much room to improve in English

---

## Round 0.2 · accepted · Accept

The authors have properly addressed all the issues suggested by reviewers.

Reviewer 1 ·

Basic reporting

After revision, the manuscript has been improved significantly and comprehensively including clear English description, professional article structure and detailed description of figs and results, all of whih are in line with the standards of PeerJ.

Experimental design

Experimental design is well-done and rigorous investigation was performed, the results are solid and convincing.

Validity of the findings

These findings are convincing.

Additional comments

All the comments have been addressed by you. Thanks. The revised manuscript can be considered for acceptance.

Reviewer 2 ·

Basic reporting

Based on prior suggestions from reviewers, the authors have made necessary changes and answered practically all the questions point-by point. Minor mistakes including nomenclature in biliary dysplasia and grading has been corrected. Some issues associated study design and methodology still exist, but overall it does affect the conclusion of this study. The title of the manuscript has been revised for easy understanding and the English of the entire manuscript has been significantly polished.

Overall, the authors reply satisfies reviewers' critique and the manuscript is ready to be accepted.

Experimental design

as stated before, the study is overall well-designed.

Validity of the findings

the study included large case series with good follow up. The immunohistochemical data from TMA also well-analysed.

Additional comments

thanks for the great effort from the authors. The overall writing has been significantly improved and minor mistakes has been corrected or answered. Additional data on resection margins and tumor stages have been provided.